# Pain Education in the Wellness, Training Performance, and Pain Intensity of Youth Athletes: An Experimental Study

**DOI:** 10.3390/healthcare12020215

**Published:** 2024-01-16

**Authors:** Andreu Sastre-Munar, Natalia Romero-Franco

**Affiliations:** 1Nursing and Physiotherapy Department, University of the Balearic Islands, 07122 Palma de Mallorca, Spain; a.sastre@uib.es; 2Sport High Performance Centre of Balearic Islands, 07009 Palma de Mallorca, Spain; 3Health Research Institute of the Balearic Islands (IdISBa), 07120 Palma de Mallorca, Spain

**Keywords:** sport, athletes, pain, wellness, education

## Abstract

Background: Although pain management programs reduce pain and improve wellness perception in the general population, few studies have explored these effects in athletes. This study evaluated the effects of an educational program about pain neuroscience on wellness, training performance, and pain in youth athletes. Differences according to sex were also explored. Methods: For 12 weeks, 52 athletes were randomly assigned to an intervention group (IG: educational program about healthy sports habits and pain neuroscience) or a control group (CG: education on healthy sports habits only). Before the start of the study and weekly until its end, wellness, training performance, and pain intensity were monitored via a questionnaire. Results: After the intervention, IG decreased stress (*p* = 0.028) compared to the baseline, and a higher number of training sessions were performed without health problems (76.6%) compared to the number in the CG (63.0%) (χ^2^ = 8.31, *p* = 0.004). Regarding pain, the IG perceived lower pain than the CG did (*p* = 0.028). Females in the IG had lower pain than those in the CG did (*p* < 0.05), without differences in other variables or in males (*p* > 0.05). Conclusions: An educational program that includes pain neuroscience may help youth athletes improve their wellness status, pain intensity perception, and training session performance.

## 1. Introduction

Maximal sports performance requires great physical and cognitive effort from athletes, who have to cope with stressful situations during training. Due to the high prevalence of injuries in sports, athletes tend to normalize the pain experienced during sports practice. Studies that have explored pain prevalence in the sports context have reported that 20–82.9% of athletes have pain [1,2,3,4].

As options for managing pain, previous studies have suggested cognitive behavioral therapy and multidisciplinary pain programs [5]. These educational therapies have shown benefits for reducing musculoskeletal pain and improving physical performance, wellness perception, and psychosocial factors in different populations with chronic pathologies [5,6].

In sports, these educational strategies have been explored to reduce pain perception only in the case of injured athletes [7,8,9]. Very few studies to date have evaluated the effects of pain management programs for informing uninjured athletes about their pain experiences. According to previous studies, athletes—especially those of younger ages—and coaches tend to normalize pain during sports practice and consider pain a necessary component during training to improve sports performance [8,10,11]. However, the frequent existence of pain is considered a potential factor in developing an injury, which ends up affecting the wellness perception and sports performance of athletes [12]. Additionally, several studies have demonstrated a positive association between training volume and pain [2,13]. The literature also suggests that female athletes may experience higher pain intensity than male athletes due to a greater fear of physical damage or a catastrophic disposition in the sporting context [14].

Furthermore, among the attributes of sport, pain is negatively associated with both wellbeing and training volume. Thus, both factors need to be monitored concurrently [15]. However, there is a dearth of research exploring the interplay among the characteristics of sport, training intensity, and pain. Despite the aforementioned arguments, there are not enough studies exploring the effects of educational programs for managing pain that are designed with an explanation founded in pain neuroscience and are focused on the specific demands of the sports population. For this reason, this study aimed to evaluate the effects of an educational program about pain neuroscience on wellness status, sports performance, and pain intensity perception in youth athletes. Taking into account the potential differences regarding the pain experience according to sex [14,16], differences in female and male athletes were also explored. We hypothesize that athletes who receive an educational program that includes an explanation founded in pain neuroscience will have improved wellness status and decreased pain intensity. Furthermore, female athletes will be more prone to showing these improvements.

## 2. Materials and Methods

### 2.1. Design

A parallel two-group randomized trial was designed. Over 12 weeks, all athletes received an educational program. The intervention group (IG) received an educational program consisting of two parts: Part 1 contained information on healthy habits during sports practice (rest, nutrition, body care, and recovery); Part 2 contained information related to pain neuroscience (biological, psychological, and perceptual aspects of pain in the sports context). The control group (CG) only received Part 1 of the program. This study was carried out in the High-Performance Sports Center of the Balearic Islands from February to May 2023. At the start of the study and weekly until its end, wellness status, training session performance, and pain intensity were monitored using a smartphone application (SaluTrack version 1.0.5) specifically designed for this purpose [17].

### 2.2. Participants

The sample size was calculated using the GRANMO application, Version 7.12 (Spain). Taking a 5% significance level and 80% statistical power, 17 athletes were required for the IG and 17 for the CG to recognize a statistically significant difference greater than or equal to 2.5 points in pain intensity and 5 points in wellness perception. The common standard deviation (SD) was assumed to be 1.7 points in pain intensity and 3.9 points in wellness perception, similarly to previous studies [18,19]. A dropout rate of 20% was anticipated. With regard to eligibility criteria, the athletes needed to be at least 14 years old and to have 2 years of sports experience to participate in national-level competitions, not to have sustained any injury or any surgery in the last 6 or 12 months, respectively, prior to the beginning of the study, and not to have a chronic disease that could influence muscle pain (i.e., knee osteoarthritis). All athletes from the aforementioned sports center were invited to participate in the study via extensive mailing from their regional sports federation. Of the 62 athletes enrolled, 60 were selected, and 52 athletes finished the study (Figure 1). The training routine was similar for all athletes because they belonged to the same high-performance program in the center. Their training components held a dual focus encompassing both technical–tactical and physical conditioning training. All athletes performed eight or nine training sessions per week. The characteristics of all participants are included in Table 1.

Prior to the beginning of the study, all participants—or their legal tutors or parents in the case of minors—gave their informed consent. This study was approved by the ethical committee of the local university (Ref. no.: 280CER22; Date: 14 July 2022). The trial was prospectively registered on ClinicalTrials.gov (ID: NCT05645562).

### 2.3. Procedures

One week before starting the educational program, all athletes were evaluated to obtain their body mass and height using a ±100 g precision digital weight scale (Tefal, France) and a t201-t4 adult height scale (Asimed, Spain), respectively. All measurements were taken in the morning (10–12 a.m.), with athletes wearing only underwear and no shoes. In this same session, all athletes completed an online questionnaire to collect the following information: (1) sociodemographic and sports data (age, sex, sports experience, and history of previous injuries); (2) wellness perception; (3) training session performance; (4) pain intensity. As wellness, training sessions, and pain intensity were monitored weekly, a smartphone application (SaluTrack) specifically designed to monitor these variables was used [17]. The athletes received an informative session on registering in the application and on how to use it.

#### 2.3.1. Smartphone Application (SaluTrack)

This smartphone application was designed for both Android and iOS devices to evaluate and monitor the training load and the psycho-physiological and physical parameters of the athletes. In our study, data collection was carried out using three questionnaires: wellness, training performance, and pain intensity. The application was designed to send weekly pop-up notifications asking athletes to complete the questionnaires [17].

#### 2.3.2. Wellness Questionnaire

Four questions assessed the quality of sleep, amount of stress, level of perceived fatigue, and perceived muscle soreness. Based on previous studies [20], each item was individually scored from 1 (“Very, very low or very, very good”) to 7 (“Very, very high or very, very bad”), with 28 being the maximal score (the worst wellness status perception). Additionally, the athletes were asked to report the effort that they perceived when training as a variable regarding the intensity of the training load [21], and this was evaluated using the 10-point Borg Rating of Perceived Exertion (RPE) scale (0 = no effort; 10 = maximum possible effort) [22].

#### 2.3.3. Training Performance

The number of training hours was registered weekly by the athletes on their smartphone application. The athletes were also asked to report on the training sessions with or without health problems using one of the following options: (a) full participation without health problems; (b) full participation with health problems; (c) reduced participation due to health problems; (d) could not participate due to health problems. This procedure was recommended by previous studies [4,23].

#### 2.3.4. Pain Intensity Questionnaire

Pain intensity was assessed using the Visual Analog Scale (VAS), ranging from 0 (no pain) to 10 (the worst imaginable pain) [24].

#### 2.3.5. Educational Program

The educational program consisted of four sessions led by experienced sports health professionals. All sessions were in groups of 10–15 people with visual support (Microsoft PowerPoint version 2312, USA); they lasted 20 min in the CG and 30–40 min in the IG. The first session took place at the onset of the study, and subsequent sessions were conducted every three weeks until the end of the study (Figure 2).

For the CG, the educational sessions provided information and self-care advice on the importance of rest, dietary habits, recovery techniques, and the principles of training based on similar studies focused on athletes [25,26] (Figure 2).

For the IG, apart from the information provided for the CG, athletes received an explanation of pain neuroscience based on previous studies [27,28,29,30,31,32] (Figure 2).

### 2.4. Statistical Analysis

Descriptive data are presented as the mean and standard deviation (SD) for numerical variables and as percentages for categorical variables. The normality of the data was evaluated using the Shapiro–Wilk test. The baseline characteristics of the athletes were compared using independent-sample Student *t*-tests. Mann–Whitney U test non-parametric analysis was used weekly to evaluate the differences between groups in the pain, wellness, and training variables during the study, and Friedman rank two-way analysis of variance was used to find weekly intragroup differences. The 95% confidence interval (95%CI) was calculated for all differences, and Cohen’s effect sizes (ES) were obtained and interpreted as follows: small (*d* ≤ 0.2), moderate (0.2 > *d* ≤ 0.8), or large (*d* > 0.8) [33]. The Chi-square (χ^2^) was used to find differences between groups in athletes who completely stopped their training sessions and those who trained with musculoskeletal problems [34]. Pearson’s correlation was used to find a relation among the wellness, intensity of pain, and training performance variables, with interpretations according to the following thresholds: small (*r* = 0.1), moderate (*r* = 0.3), large (*r* = 0.5), very large (*r* = 0.7), and extremely large (*r* = 0.9) [35]. International Business Machines (IBM) SPSS Statistics, Version 21.0 (Chicago, IL, USA) was used, and statistical significance was set at *p* < 0.05.

## 3. Results

The baseline demographic, sport, pain, and wellness data did not show any statistical differences between the CG and the IG (*p* > 0.05) (Table 1 and Table 2).

### 3.1. Wellness

Regarding between-group differences, we observed lower values for the IG than for the CG in muscle soreness at the 8th (*p* = 0.033, *d* = 0.77) and 10th (*p* = 0.040, *d* = 0.91) weeks (Appendix A). Within-group differences showed that the IG perceived lower stress than at the baseline (*p* = 0.028, *d* = 0.51). No other significant differences were found for sleep, stress, fatigue, muscle soreness, or total wellness (*p* > 0.05) (Table 2). Furthermore, no differences in RPE levels were found (*p* > 0.05).

When exploring these results according to sex, female athletes in the IG had lower muscle soreness values than those in the CG did at the 7th (*p* = 0.023, *d* = 1.0), 8th (*p* = 0.006, *d* = 1.43), 10th (*p* = 0.019, *d* = 0.91), and 11th (*p* = 0.008, *d* = 2.07) weeks. Also, fatigue was higher in females from the IG than in females from the CG at the 5th week (mean difference = −0.93 ± 0.45, *p* = 0.036, *d* = 0.82, 95%CI = −1.86 to −0.01). The total wellness score was lower in females from the IG (9.73 ± 5.00) than in females from the CG (13.92 ± 2.35) at the 10th week (*p* = 0.037, *d* = 1.07, 95%CI = 0.85–7.53). Within-group differences showed that females in the IG perceived lower stress at the end of the educational program than they did at the baseline (*p* = 0.046, *d* = 0.57) (Appendix A). No significant differences were shown for male athletes (*p* > 0.05) (Appendix A).

### 3.2. Training Performance

Significant between-group differences were found, with athletes in the IG completing more full training sessions without health problems (76.6%) than athletes from the CG (63.0%) (χ^2^ = 8.31, *p* = 0.004, valor = 8.31). Thus, athletes in the IG carried out more training hours compared to athletes in the CG at the 3rd (*p* = 0.27, *d* = 0.76) and 9th (*p* = 0.29, *d* = 2.68) weeks. A total of 48.9% of athletes from the CG and 22.5% from the IG could not participate in their training sessions due to health problems; this difference was statistically significant (χ^2^ = 6.37, *p* = 0.027). No differences were found when exploring data according to sex (*p* > 0.05).

### 3.3. Pain Intensity

Regarding between-group differences, athletes in the IG perceived less pain intensity than athletes in the CG at the 2nd (*p* = 0.031, *d* = 1.19), 5th (*p* = 0.029, *d* = 2.32), and 10th (*p* = 0.029, *d* = 2.33) weeks. Athletes in the IG also perceived lower pain intensity than athletes in the CG did after the intervention (*p* = 0.028, *d* = 1.16) (Table 2).

When exploring pain intensity according to sex, female athletes in the IG had lower values than those of female athletes in the CG at the 2nd (*p* = 0.38, *d* = 1.96) and 10th (*p* = 0.29, *d* = 2.60) weeks. Also, the IG had a lower pain level than that of the CG after the intervention (*p* = 0.42, *d* = 0.41) (Appendix A). No significant differences were observed for male athletes (*p* > 0.05) (Appendix A).

### 3.4. Correlations

Pearson’s correlation analysis revealed significant associations between wellness and several factors. Higher wellness scores were positively correlated with increased training hours (*p* < 0.01, *r* = 0.23), a greater number of competitions (*p* < 0.05, *r* = 0.24), and elevated intensity of pain (*p* = 0.02, *r* = 0.34).

## 4. Discussion

The main findings of the present study showed that a 12-week educational program that included information related to pain neuroscience reduced the stress and pain intensity of athletes, and it enabled them to perform more training sessions and more hours of training without health problems compared with athletes who did not receive the pain neuroscience part of the program.

Regarding perceived stress (one item of wellness status), the athletes in the IG showed a lower level of stress than the athletes in the CG did after the 12-week period of the intervention. This was in line with the results reported by Louw et al. in 2011 [36], as they reported the beneficial effects of pain education on anxiety and stress in a population with musculoskeletal pain.

With regard to muscle soreness (another item of wellness status), only athletes in the IG had improved values compared to the baseline, even though the athletes in the CG also received educational information related to healthy habits in sports. This is in contrast with previous studies showing that programs on healthy habits in sports produced beneficial effects on muscle soreness in athletes [37,38]. This difference might be because this study conducted an educational session on recovery techniques, whereas other studies implemented these techniques as practical content [37,38].

We should highlight the lack of effects on sleep and fatigue in the athletes despite their receiving an educational program related to healthy habits in sports. These results disagree with those of previous studies that observed higher hours of sleep, sports performance, and the existence of nutritional habits after an educational program with content related to nutrition [25] or sleep habits [26]. As an explanation, these studies focused their educational programs on only one of these topics (nutrition or sleep), whereas our program addressed different topics, with only one session for each.

In terms of training performance, we observed that athletes in the IG were able to complete more hours of training than athletes in the CG. This finding, along with the reductions in stress and pain intensity observed in the athletes in the IG, may mean that increasing the information on the neurological mechanisms of pain may help athletes recognize and monitor stressful situations during training and complete their sessions [31,32]. This suggestion is also supported by the fact that the athletes in the CG displayed a higher number of training sessions with health problems compared to the athletes in the IG, and this is aligned with the literature suggesting that athletes may normalize pain to pursue their goals [8,10,11]. To confirm this explanation, more studies exploring the qualitative perception of athletes are needed.

Regarding pain perception, the athletes in the IG reported a lower level of pain than the athletes in the CG did during the 12-week intervention period and thereafter. These findings agree with those of similar studies that administered pain neuroscience education in a population with chronic musculoskeletal pain [28,39]. With regard to the sports population, athletes always achieved better pain management and even a lower fear of movement [7,29]. However, these studies always administered pain neuroscience programs among injured athletes. Due to the importance of pain in the sports context, even for healthy athletes, future educational programs should be designed and incorporated into the sports population.

In relation to pain intensity according to gender, our findings revealed that only female athletes in the IG had a reduction in pain intensity compared to females in the CG during and after the 12-week intervention period. To the best of our knowledge, no other studies have examined the sex differences in pain following an educational program. Considering that the intervention was an educational program that was similar for both female and male athletes, this discrepancy could be attributed to sex-specific coping mechanisms. In this sense, previous studies have demonstrated that males typically resort to behavioral distraction and problem-focused strategies to cope with pain, whereas females often use a broader array of coping techniques, such as seeking social support, utilizing positive self-statements, employing emotion-focused strategies, and engaging in cognitive reinterpretation [16,40]. Additionally, societal expectations associated with gender roles play a part in this pattern. The stereotypical feminine role is linked with a greater inclination to openly report pain in the general population, whereas the expected masculine role is more commonly associated with stoicism [40]. Furthermore, a study conducted by Diotaiuti et al. in 2022 [14] demonstrated that female athletes experienced higher pain intensity and a lower pain threshold than male athletes did. This explanation could also explain the absence of differences in pain intensity among male athletes in the present study.

Similarly, in terms of wellness, the effects of the intervention focused on pain neuroscience education were only observed in female athletes. In this regard, although some of the scientific literature has analyzed sex-specific differences in subjective wellness, very few studies to date have explored them in the sports context. Although previous studies suggested hormones as an important influencing factor [41], the aforementioned social context should also be considered. Additionally, our results showed that higher levels of wellness were positively correlated with pain intensity, training hours, and number of competitions. This is in line with a study by Dudley et al. from 2023 [42], which suggested that a high training volume might be associated with the health of athletes, increasing the risk of injury and illness.

The present study had several limitations. Firstly, the specific composition of the sample (young athletes from a high-performance center) may affect the generalizability of the results to other athlete populations. Secondly, although our athletes belonged to a high-performance center that requires participation in at least national-level competition, some variations in individual performance levels may exist. As higher levels of sports performance may be associated with more positive mental health [43], this aspect could have affected our results. Furthermore, in our study, sessions of the educational program were administered in groups of 10–15 athletes. Although the scientific literature recommends face-to-face sessions [6,36], this is not possible in sports organizations with a large number of athletes. Thus, the organization of grouped sessions helped to maintain the viability of our study. Finally, we used a self-administered questionnaire to evaluate and monitor all data. Due to the subjectivity of wellness status and pain perception, qualitative aspects of these results need to be explored in future studies to gain a more complete understanding.

As practical applications, health and sports professionals should consider that an increase in information related to the mechanisms of pain neuroscience may help young athletes reduce their perception of stress, pain, and muscle soreness. These achievements may make them able to complete more hours of training with fewer health problems.

For future studies, we suggest that qualitative methodologies should be incorporated into further research to better understand the pain perceptions of athletes and the impacts of educational programs. This approach could have the potential to yield a deeper understanding and contribute significantly to the design of educational programs and therapies in this field. Although the program addressed several topics, such as the neuroscience of pain, the lack of significant effects on sleep and fatigue may reflect that these aspects should be specifically addressed. We suggest exploring how to address these aspects more effectively in future research. Also, future studies should consider the diversity of sports populations and sex when designing educational programs to determine their applicability and effectiveness in different contexts.

## 5. Conclusions

In conclusion, a 12-week educational program that includes information on pain neuroscience may help athletes reduce stress and pain intensity and enable them to complete a higher training volume compared to athletes who do not receive any pain neuroscience education.

## Figures and Tables

**Figure 1 healthcare-12-00215-f001:**
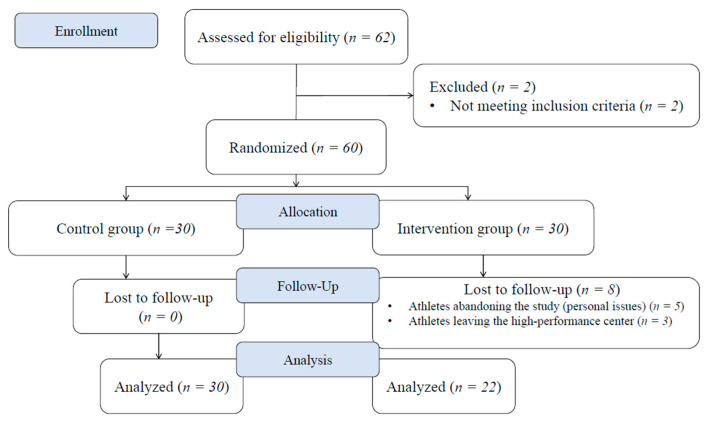
Flow diagram.

**Figure 2 healthcare-12-00215-f002:**
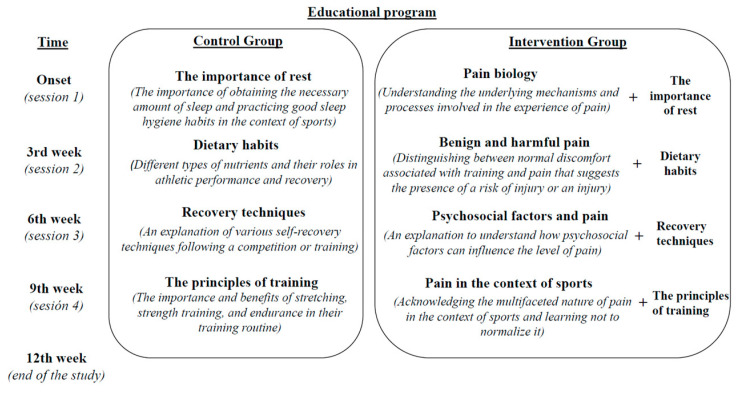
Educational program.

**Table 1 healthcare-12-00215-t001:** Demographic and sports characteristics of all participants.

	Control Group			Intervention Group		
	All Athletes (*n* = 30)	Female Athletes (*n* = 16)	Male Athletes (*n* = 14)	All Athletes (*n* = 22)	Female Athletes (*n* = 13)	Male Athletes (*n* = 9)
	Mean ± SD	Mean ± SD	Mean ± SD	Mean ± SD	Mean ± SD	Mean ± SD
Age (years)	16.00 ± 0.45	15.94 ± 0.57	16.07 ± 0.27	16.41 ± 0.73	16.31 ± 0.75	16.55 ± 0.73
Height (m)	1.75 ± 0.10	1.70 ± 0.07	1.82 ± 0.09	1.78 ± 0.08	1.75 ± 0.05	1.83 ± 0.08
Weight (kg)	68.01 ± 11.78	61.55 ± 6.32	75.54 ± 12.37	70.14 ± 9.16	65.15 ± 4.88	77.33 ± 9.27
Sports experience (years)	8.6 ± 2.97	8.1 ± 3.23	9.2 ± 2.66	9.8 ± 3.55	9.5 ± 2.82	10.3 ± 4.55
Basketball (%)	36.7	37.5	42.8	31.8	30.8	33.3
Rowing (%)	23.3	25.0	14.3	4.5	7.7	-
Rugby (%)	16.7	6.3	28.7	9.1	-	22.2
Swimming (%)	10.0	12.5	7.1	31.8	30.8	33.3
Taekwondo (%)	3.3	-	7.1	9.1	7.7	11.2
Track and field (%)		-	-	4.5	7.7	-
Volleyball (%)	10.0	18.7	-	9.1	15.3	-

SD: standard deviation.

**Table 2 healthcare-12-00215-t002:** Wellness and pain intensity of the athletes.

	Control Group				Intervention Group				Between-Group Differences	
	PRE (Mean ± SD)	POST (Mean ± SD)	Intragroup Differences		PRE (Mean ± SD)	POST (Mean ± SD)	Intragroup Differences			
			Mean (95%CI)	ES			Mean (95%CI)	ES	Mean (95%CI)	ES
Pain intensity (0–10 points)	3.67 ± 1.14	5.70 ± 2.11	−2.38 (−5.26; 0.66)	NS	5.33 ± 1.52	3.71 ± 0.95	1.50 (−4.85; 7.85)	NS	1.49 (0.11; 2.87) *	1.16
Sleep (1–7 points)	3.58 ± 1.33	3.46 ± 1.36	0.11 (−0.56; 0.79)	NS	3.23 ± 1.54	3.68 ± 1.24	−0.45 (−1.12; 0.21)	NS	−0.22 (0.38; −0.98)	NS
Stress (1–7 points)	3.69 ± 1.57	3.65 ± 1.29	0.38 (−0.50; 0.58)	NS	4.31 ± 1.58	3.54 ± 1.40	0.77 (0.76; 1.47) *	0.51	0.11 (0.39; −0.68)	NS
Fatigue (1–7 points)	4.46 ± 1.30	4.04 ± 1.14	0.42 (−0.05; 0.89)	NS	4.31 ± 1.25	4.22 ± 0.97	0.09 (−0.61; 1.16)	NS	−0.19 (0.31; −0.81)	NS
Muscle soreness (1–7 points)	3.77 ± 1.47	4.23 ± 1.33	−0.46 (−1.15; 0.23)	NS	4.00 ± 1.48	3.72 ± 1.31	0.27 (−0.61; 1.16)	NS	0.50 (0.38; −0.27)	NS
Total wellness (7–28 points)	15.50 ± 3.54	15.38 ± 3.06	0.11 (−1.14; 1.37)	NS	15.86 ± 2.93	15.18 ± 3.12	0.68 (−0.77; 2.14)	NS	0.20 (0.89; −1.60)	NS

CI, confidence interval; ES, effect size; NS, non-significant; SD, standard deviation; * *p* < 0.05.

## Data Availability

The data presented in this study are available on request from the corresponding author. The data are not publicly available due to privacy.

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
