# Peer review of "Pain Education in the Wellness, Training Performance, and Pain Intensity of Youth Athletes: An Experimental Study"

_healthcare, 2024, doi:10.3390/healthcare12020215_

Round 1

Reviewer 1 Report

Comments and Suggestions for Authors

Dear authors

Article is properly written and focused on an interesting topic.  Only few things need to be revised to make it clearer.

-Line 150-151: demographic data including mean age, gender and other information should be included in material and methods instead of results paragraph, please correct it.

-Your study has been approved by your Ethical committee , could be useful to clarify that informed consent has been obtained from all the participants and if participants were under the minor legal age informed consent was signed by their parents.

Thank you for submitting at Healthcare. We’re looking forward to receive your update paper. 

Author Response

Dear Reviewer:

Reviewer 2 Report

Comments and Suggestions for Authors

Article: Pain education in wellness, training performance and pain intensity of youth athletes: an experimental study

Dear authors,

The study is interesting, however I offer below some suggestions that you can consider so that the comparison between groups and conclusions are more accurate.

We need to know other variables, e.g. if any participant has a chronic disease, such as knee osteoarthritis, which could influence muscle pain.

In addition to the experience mentioned (years of practice), it is necessary to know the level of trainability of each participant. Was there a prior assessment of the training status (performance level) of each participant?

In the training performance dimension, the focus is on volume and frequency, although it is necessary to make reference to the level of intensity (a crucial component of the training load for sports performance).

What are the fundamental aspects of the training load (nature, magnitude, orientation) and the objectives of the training sessions carried out? Were the participants subjected to technical-tactical or physical conditioning training?

Kind regards

Author Response

Dear Reviewer:

Reviewer 3 Report

Comments and Suggestions for Authors

The study presents interesting findings on the effects of a 12-week educational program incorporating pain neuroscience content in young high-performance athletes. Here are some critical comments and reflections on the article:

Positive aspects:

Positive impact on overall well-being: The results show that the educational program had a positive impact on reducing stress and pain intensity in athletes, which may be beneficial to their overall well-being.

Improved performance: It is encouraging to note that athletes in the group that received pain neuroscience education were able to perform more hours of training without experiencing health problems compared to the control group.

Gender differences: The study addresses gender differences in pain perception, highlighting the reduction in pain intensity in female athletes who received pain neuroscience education.

Aspects to consider:

Limitations of the study: some limitations are mentioned, such as the specific composition of the sample (young athletes from a high performance center), which may affect the generalizability of the results to other athlete populations.

Assessment method: The study uses self-administered questionnaires to assess well-being and pain perception, which introduces a subjective component. It is suggested that qualitative aspects need to be explored to gain a more complete understanding.

Comprehensive approach: Although the program addresses several topics, such as the neuroscience of pain, the lack of significant effects on sleep and fatigue could be an area for improvement. We suggest exploring in future research how to address these aspects more effectively.

Differences with previous studies: The article points out differences in the results compared to previous studies on education in healthy habits in athletes. It is hypothesized that the difference may be due to the inclusion of specific educational sessions on recovery techniques in this study.

Recommendations for future research:

Qualitative exploration: It is suggested that qualitative methodologies be incorporated in future research to better understand athletes' perception of pain and the impact of the educational program.

Generalization of results: Consider the diversity of sport populations when designing educational programs to determine applicability and effectiveness in different contexts.

Gender-specific approach: Given that gender differences in pain perception were observed, future research could further explore how to tailor educational programs to address the specific needs of male and female athletes.

In summary, the study provides valuable insight into the potential benefits of pain neuroscience education in young athletes, but also points to areas for future research and methodological improvements.

Author Response

Dear Reviewer:

Reviewer 4 Report

Comments and Suggestions for Authors

Dear Authors,

please see the attached file. 

Author Response

Dear Reviewer:

Reviewer 5 Report

Comments and Suggestions for Authors

The study adds interesting information that could be useful for people who approach to sports to improve the performance. However, some points need to be improved and explained. Below, my comments and indications:·      Lines 15 and 19: please, remove the semicolon and insert the dot at the end of the paragraph in each subsection.

·      Line 48-50: the authors should report some studies, currently available, that show pain in different sports and the association with both well-being and training volume and such differences according to gender.

·      Line 85: the authors states that 56 athletes were screened and 54 were enrolled while the figure 1 reports 62 eligible and 60 took part to study, please clarify this point.

·      Line 86: the authors stated that “The training routine was similar for all athletes…”, it concerning the volume in terms of number of sessions per week? this point should be better clarified.

·      Line 92-94: please, also indicate the date of ethical committee approval.

·      Table 1: please, report plus/minus (±) for SD.

·      Table 1: “years” is reported in long form and then in short form, please standardize.

·      Line 99: please, indicate the procedure to measure weight and height (e.g. time of day, with/without clothes etc)

·      Line 141: it should be figure 2, please check.

·      Line 149 and 151: as above.

·      Line 167: are significant differences between age, weight, and height between groups? The authors should indicate these data in results section.

·      Line 180: as indicated for line 141 (see above).

·      Table 2: please, report plus/minus (±) for SD.

·      Line 267-272: the authors, as already suggested for the introduction (see above), should report the main results of the studies in which sex-specific differences regarding the parameters analyzed in the present manuscript and try to discuss.

·      Line 273: although the limitations of the study are supportable, I was thinking why the authors did not also include physical performance levels measured by indicators (e.g. cardiorespiratory capacity) in the evaluations. Moreover, the authors should argue and explain more extensively the results performing add further statistical investigations such as correlations between variables detected.

·      Line 317-318: please, also indicate the date of ethical committee approval.

·      Line 326: please, report references according to journal guidelines. Some references the DOI are misses while other report it, please standardize all.

Comments on the Quality of English Language

The manuscript needs to be reviewed for the English language by a native speaker

Author Response

Dear Reviewer,

Thank you for your helpful comments and your proposal to resubmit our paper entitled: “Pain education in wellness, training performance and pain intensity of youth athletes: an experimental study”, pending the correction of these some major issues.

Authors are very grateful for the useful comments and time spent. The modifications have been marked up using the “Track Changes” function such that any changes can be easily located. Also, the replies to reviewer' comments are included, point by point, by explaining the details of the revisions, in bold format.

We thank you for your consideration.

Yours sincerely,

The authors

The study adds interesting information that could be useful for people who approach to sports to improve the performance. However, some points need to be improved and explained. Below, my comments and indications:·     

We thank this supportive comment.

 Lines 15 and 19: please, remove the semicolon and insert the dot at the end of the paragraph in each subsection.

We have corrected these punctuations: lines 15-19.

  • Line 48-50: the authors should report some studies, currently available, that show pain in different sports and the association with both well-being and training volume and such differences according to gender.

According to the reviewer recommendation, these aspects regarding gender differences have been extended in the introduction section: lines 46-50.

  • Line 85: the authors states that 56 athletes were screened and 54 were enrolled while the figure 1 reports 62 eligible and 60 took part to study, please clarify this point.

We have corrected this issue in the text: lines 88-89.

  • Line 86: the authors stated that “The training routine was similar for all athletes…”, it concerning the volume in terms of number of sessions per week? this point should be better clarified.

We agree with the reviewer and thus, we have extended this information to clarify this point. Line: 93

  • Line 92-94: please, also indicate the date of ethical committee approval.

Added: Line 99

  • Table 1: please, report plus/minus (±) for SD.

Added to the Table 1 and Table 2.

  • Table 1: “years” is reported in long form and then in short form, please standardize.

We have standardized this term in Table 1.

  • Line 99: please, indicate the procedure to measure weight and height (e.g. time of day, with/without clothes etc)

We agree with the reviewer. This procedure has been extended in the Materials and Methods section: lines 108-109.

  • Line 141: it should be figure 2, please check.

We agree with the reviewer. We have corrected this mistake: line 148.

  • Line 149 and 151: as above.

Corrected.

  • Line 167: are significant differences between age, weight, and height between groups? The authors should indicate these data in results section.

We agree with the reviewer. These results have been mentioned in Results section: lines 176-177.

  • Line 180: as indicated for line 141 (see above).

In this case, we are referring to the Figure S1 (included as supplemental file).

  • Table 2: please, report plus/minus (±) for SD.

Added.

  • Line 267-272: the authors, as already suggested for the introduction (see above), should report the main results of the studies in which sex-specific differences regarding the parameters analyzed in the present manuscript and try to discuss.

According to the reviewer recommendation, this information has been added: lines: 260-262 and 271-273.

  • Line 273: although the limitations of the study are supportable, I was thinking why the authors did not also include physical performance levels measured by indicators (e.g. cardiorespiratory capacity) in the evaluations. Moreover, the authors should argue and explain more extensively the results performing add further statistical investigations such as correlations between variables detected.

According to the reviewer recommendation, correlations between variables were detected and included. This information was added in Materials and Methods section (lines 170-173), Results section (lines 212-216) and Discusion section (lines 281-285).

Regarding the proposal to include physical performance levels, such as cardiorespiratory capacity, we would like to clarify that the primary aim of our study does not involve assessing these aspects. Instead, we endeavored to gauge the performance level by integrating the competitive level.

  • Line 317-318: please, also indicate the date of ethical committee approval.

Added: line 327.

  • Line 326: please, report references according to journal guidelines. Some references the DOI are misses while other report it, please standardize all.

Upon considering the recommendations of the reviewers and recognizing that the DOI for some references were unavailable, we opted to standardize all references without DOI.

Comments on the Quality of English Language

The manuscript needs to be reviewed for the English language by a native speaker

According to the recommendation of the reviewer, our manuscript has been proofread by a professional service.

Round 2

Reviewer 4 Report

Comments and Suggestions for Authors

Dear Authors,

Thank you for your effort. Now, your manuscript has been substantially improved. Congratulations

Author Response

Authors are very grateful for the supportive comments and time spent once again. We thank you for your consideration.

Reviewer 5 Report

Comments and Suggestions for Authors

Dear Authors, I would like to express my gratitude for considering my suggestions in order to improve the manuscript. Below just a few other observations:

Line 216: the header of the sub-paragraph should be replaced with "3.4 Correlations"

References: please, report references according to journal guidelines. For instance regarding journal articles as follow: 

Author 1, A.B.; Author 2, C.D. Title of the article. Abbreviated Journal Name Year, Volume, page range.

Fai

Author Response

Dear Reviewer:

Once again, thank you for your new comments and your proposal to resubmit our paper “Pain education in wellness, training performance and pain in-tensity of youth athletes: An experimental study”, pending the correction of these some minor issues.

Authors are very grateful for the useful comments and time spent. The modifications have been marked up using the “Track Changes” function such that any changes can be easily located. Also, the replies to reviewer' comments are included, point by point, by explaining the details of the revisions, in bold format, in the attached document.

We thank you for your consideration and hope that our responses will come up to your expectations.

Yours sincerely,

Dear Authors, I would like to express my gratitude for considering my suggestions in order to improve the manuscript. Below just a few other observations:

Line 216: the header of the sub-paragraph should be replaced with "3.4 Correlations"

Replaced.

References: please, report references according to journal guidelines. For instance regarding journal articles as follow: 

Author 1, A.B.; Author 2, C.D. Title of the article. Abbreviated Journal Name YearVolume, page range.

All the references have been checked according to journal guidelines.